# Extended Abstract Track

# Object-Centric Causal Representation Learning

**Amin Mansouri**            AMIN.MANSOURI@MILA.QUEBEC
**Jason Hartford**            JASON.HARTFORD@MILA.QUEBEC
**Kartik Ahuja**            KARTIK.AHUJA@MILA.QUEBEC
**Yoshua Bengio**            YOSHUA.BENGIO@MILA.QUEBEC
*Mila, Quebec AI Institute*

**Editors:** Sophia Sanborn, Christian Shewmake, Simone Azeglio, Arianna Di Bernardo, Nina Miolane

## Abstract

There has been significant recent progress in causal representation learning that has showed a variety of settings in which we can disentangle latent variables with identifiability guarantees (up to some reasonable equivalence class). Common to all of these approaches is the assumption that (1) the latent variables are $d-$dimensional vectors, and (2) that the observations are the output of some injective observation function of these latent variables. While these assumptions appear benign—they amount to assuming that any changes in the latent space are reflected in the observation space, and that we can use standard encoders to infer the latent variables—we show that when the observations are of multiple objects, the observation function is no longer injective, and disentanglement fails in practice. We can address this failure by combining recent developments in object-centric learning and causal representation learning. By modifying the Slot Attention architecture (Locatello et al., 2020b), we develop an object-centric architecture that leverages weak supervision from sparse perturbations to disentangle each object's properties. We argue that this approach is more data-efficient in the sense that it requires significantly fewer perturbations than a comparable approach that encodes to a Euclidean space and, we show that this approach successfully disentangles the properties of a set of objects in a series of simple image-based disentanglement experiments.

**Keywords:** Disentanglement, object-centric learning, causal representation learning

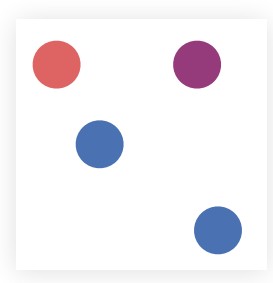 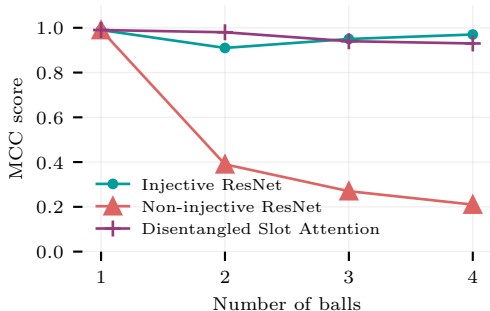

Figure 1: *(Left)* An example image of simple objects. *(Right)* Objects result in injectivity failures which cause disentanglement procedures to fail.

Extended Abstract Track

## 1. Introduction

Consider the image in Figure 1 (left). We can clearly see four different colored balls, each at a different position. But asking, "Which is the first shape? And which is the second?" does not have a clear answer: the image just depicts an unordered set of objects. This observation seems trivial, but it implies that there exist permutations of the objects which leave the image unchanged. For example, we could swap the positions of the two blue balls without changing a pixel in the image.

In causal representation learning, the standard assumption is that our observations, $x$, are "rendered" by some generative function $g(.)$ that maps the latent properties of the image, $z$, to pixel space (i.e. $x = g(z)$); the goal is to *disentangle* the image by finding an "inverse" map that recovers $z$ from $x$ up to some irrelevant transformation. The only constraint on $g(\cdot)$ that is assumed by all recent papers (for example Hyvarinen and Morioka, 2016, 2017; Locatello et al., 2020a; Khemakhem et al., 2020a,b; Lachapelle et al., 2022; Ahuja et al., 2022a,b), is that $g(\cdot)$ is injective, such that $g(z_1) = g(z_2)$ implies that $z_1 = z_2$. But notice that if we represent, $z \in \mathbb{R}^d$, then whenever we observe objects like those shown in Figure 1, this injectivity assumption fails because symmetries in the objects' pixel representation imply that there exist non-trivial permutation matrices $P$, such that $g(z) = g(Pz)$. This is not just a theoretical inconvenience: Figure 1 (right) shows that when the balls are indistinguishable, the disentanglement performance of a recent approach from Ahuja et al. (2022b) is upper-bounded by $1/k$ where $k$ is the number of balls.

The key to avoiding this problem, is to recognize that the latent representations of multi-object images are sets, and should be treated as such by our encoders in order to enforce invariance among these permutations. Of course, this requires that we have an encoder that can partition an image into a set of latents, instead of the typical single latent vector that would be output by a standard encoder architecture. Fortunately, the task of segmenting an image into its constituent objects has been extensively studied in object-centric learning (see e.g. Bapst et al., 2019; Locatello et al., 2020b; Goyal et al., 2020; Greff et al., 2020), so there now exist a variety of architectures that can be leveraged for this task. In this work, we use Locatello et al.'s slot attention architecture and adapt it to disentangle the properties of each object using weak supervision in the style of Ahuja et al. (2022a,b).

Slot attention is usually trained with a reconstruction loss from a relatively high-dimensional per-object representation, but for the images that we work with, we want a relatively low dimensional latent description (in the simplest case, just a two dimensional position vector for each object). To disentangle these high-dimensional slot representations, we simply add a projection head that is weakly supervised by a latent space loss. Ahuja et al. use either knowledge of the mechanisms that govern the dynamics of the system, or sparse transitions between images to define this disentanglement loss. In both cases, switching to a slot attention architecture requires a matching step to infer the object to which the mechanism or offset was applied. When the mechanism is known, we use either bipartite matching using the Hungarian algorithm (Kuhn, 1955), or a more robust double matching procedure that requires solving a constrained linear program (See appendix B.1). The latter is robust to mechanisms that induce large visual changes in the images, or when there are multiple identical objects in a scene, but is also more computationally expensive. In settings with sparse transitions, this "matching" step reduces to a simple minimization

over a cost matrix, which is computationally cheap. We demonstrate the success of this architecture using a series of experiments on an image-based dataset that is intentionally simple to enable us to study how changes in the environment affect both slot attention and disentanglement. We show,

1. Object-centric disentanglement avoids injectivity failures and successfully recovers the object positions up to scaling and offsets.

2. The method can be applied even when the objects are visually identical, though slot attention does have more trouble separating objects into slots.

Finally we note that the pixel-level symmetries of objects result in a reduction in the number of offsets that we need to observe for a disentangled representation to be identifiable. With standard architectures, the number of offsets that we need to observe is linear in the number of objects. By contrast object-centric approaches only need as many offsets as properties that we want to disentangle, because the same encoder is used for each object.

## 2. Object-centric causal representation learning

**Data generating process** We assume a set, $Z := \{z_i\}_{i=1}^k$ of $k$ objects are drawn from some joint distribution, $\mathbb{P}_Z$. In order to compare set and vector representations, we let $\mathbf{vec}_\pi(Z)$ denote a flattened vector representation of $Z$ ordered according to some permutation $\pi \in \mathrm{Sym}(k)$, symmetric group of permutations of $k$ objects; when $\pi$ is omitted, $\mathbf{vec}(Z)$ simply refers to an arbitrary default ordering (i.e. the identity element of the group). Each object is described by a $d-$dimensional vector of properties[1] $z_i \in \mathbb{R}^d$, and hence $\mathbf{vec}(Z) \in \mathbb{R}^{kd}$. We observe images $x$ which are generated via a generative function $\mathfrak{g}(\cdot)$ that renders a set of object properties into a scene in pixel space, such that $x = \mathfrak{g}(Z)$. We assume that $\mathfrak{g}(\cdot)$ is an injective set-valued function, such that $\mathfrak{g}(Z) = \mathfrak{g}(Z')$ implies that $Z = Z'$ (i.e. set equality). While $\mathfrak{g}(\cdot)$ is a set-valued function, we can define an analogous vector-valued generative function, $g$, which, by definition, produces the same output as $\mathfrak{g}(Z)$; i.e. for all $\pi \in \mathrm{Sym}(k)$, $g(\mathbf{vec}_\pi(Z)) = \mathfrak{g}(Z)$. By construction, both functions result in the same distribution of images, but $g$ is clearly not injective because every $\pi \in \mathrm{Sym}(k)$ produces the same image.

We follow Ahuja et al. (2022b) and assume we have access to paired samples, $(x, x')$ where $x = \mathfrak{g}(Z)$, $x' = \mathfrak{g}(Z')$, and $z_i'$ is perturbed by a set of sparse offsets $\Delta = \{\delta_1, \ldots, \delta_k\}$, $\delta_i \in \mathbb{R}^d$, such that $z_i' = z_i + \delta_i$ for all $i \in \{1, \ldots, k\}$, and $\forall \delta_i$ at least one entry is zero.

**(Non-)Injective Disentanglement** In order to disentangle $x$, we want an "encoder", $f : \mathcal{X} \to \mathcal{Z}$, such that $f \circ g \circ z = z$ for all $z \in \mathcal{Z}$. It is common to treat $\mathcal{Z} = \mathbb{R}^{kd}$, so that we can approximate $f$ with a standard neural network-based encoder, such as a ResNet (He et al., 2016). If $g(\cdot)$ were injective, every $\delta \in \Delta$ is 1-sparse and we observe at least $kd$ offsets, then a minimizer of the following objective recovers the true $f$ up to permutations, scaling and an offset (Ahuja et al., 2022b, Theorem 1),

$$\hat{f} \in \arg\min_{f'} E_{x,x',\delta}\left[\left(f'(x) + \delta - f'(x')\right)^2\right] \quad \Rightarrow \quad \hat{f}(x) = \hat{z} = \Pi\Lambda\mathbf{vec}(Z) + c \qquad (1)$$

---

1. A natural extension of the perspective we take in this paper is to also treat properties as sets rather than ordered vectors. We leave that to future work.

where $\Pi$ is a permutation matrix, $\Lambda$ is an invertible diagonal matrix and $c$ is an offset.

However, as we argued above, $g(\cdot)$ is *not* injective unless there is only a single object. Because for all permutations $\pi, \pi' \in \mathrm{Sym}(k)$, $g(\mathbf{vec}_\pi(Z)) = g(\mathbf{vec}_{\pi'}(Z)) = x$, when we observe only $x$, we do not know which of the permutations of $\mathbf{vec}(Z)$ produced $x$. This results in two problems: (1) the loss function above requires that all three terms are "matched" in the sense that the perturbation, $\delta$, is applied to $f'(\cdot)$'s prediction of which of the coordinates were perturbed; because the permutations are arbitrary, this requires a two-way matching between the respective outputs of $f'$ and $\delta$. (2) If the outputs of $f$ are inferred in parallel (as in a feed-forward network), then it is not possible to coordinate a consistent correspondence between objects and outputs. E.g., if there are two objects, each with a single property such that $x = g([z_1, z_2])$, then either $\hat{f}(x) = [z_1, z_2]$ or $\hat{f}(x) = [z_2, z_1]$ would be correct up to a permutation. At the same time, there is no symmetry breaking in $x$ that prevents both outputs incorrectly conditioning on a single object, leading to two incorrect solutions, $\hat{f} = [z_1, z_1]$ or $\hat{f} = [z_2, z_2]$.

**Disentangling objects**   In order to address these issues, we use an object-centric architecture to map our observations to a set of latents. We use Locatello et al.'s slot attention architecture[2] which outputs $\{s_i\}_{i=1}^k$ slots as a function of the input image. Each of the $k$ slots uses the same architecture but conditions on a different subset of the image via an attention-based partition. They break symmetries between slots by randomly initializing each slot (which addresses issue (2) mentioned above). The architecture is trained with a reconstruction loss via a decoder from the slot representations.

We disentangle these slot vectors, $s_i$, by adding a projection architecture, $\hat{f}_s$, that maps each slot to a $d-$dimensional latent, which we supervise with Eq 1. This requires a matching step to match $\delta$ to the respective slots (issue (1) above), which we solve using either Hungarian Matching Kuhn (1955) or a Linear Program-based procedure (see Appendix B.1). If the image is correctly partitioned such that each slot only contains a single image, this effectively reduces the problem to a single object disentanglement problem in which injectivity holds. Because of this, we only need $d$ offsets to disentangle the image, instead of $kd$.

## 3. Empirical evaluation

In this section we give a very brief overview of the empirical results that compare our modified version the slot attention architecture with slot attention-based baselines as well as a ResNet18 (He et al., 2015) (denoted by ConvNet in the tables) trained on a non-injective and injective DGP[3] of 2,3,4 objects. The details of slot attention-based baselines are described below.

**Baselines**   Our method projects slots to a latent space that has the same dimensionality as the true latents. However, slot attention does not have such a projection, thus we need ways to obtain low-dimensional projections of high-dimensional slot representations so that

---

2. Given space constraints, we refer the reader to the original paper for full details of the architecture and only mention pertinent details here.

3. We make $g$ injective by uniquely coloring each object based on its order in the default permutation and keep using the same order for constructing the $\mathbb{R}^{kd}$ vector that is input to the injective $g(\cdot)$.

we can estimate their disentanglement performance. We use the following approaches to project slots to a $p-$dimensional target space:

- Random Projections (RP): Random projections preserve distances in the projected space, so we can use such projections of slot representations to obtain a crude estimate of pure slot attention's disentanglement performance.

- Principal Components (PC): We can pick the top $p$ principal components of slot representations after performing PCA analysis and compute the disentanglement scores.

- Linear Regression (LR): We can also directly use the true latents and learn a linear mapping from non-background slots to the $z$ space and use this projection for disentanglement metrics. This gives an upper bound on what slot attention could achieve under linear transformations as it is completely supervised by the knowledge of the true latents, which is far from what our method achieves; we do not exploit the ground truth properties during training.

Table 1 (in the appendix) confirms that as long as the observation function is injective we can empirically achieve identification. But the moment we drop any ordering over the objects and render $x$ via a non-injective function, then identification via ResNet18, which is suited only to injective renderers fails disastrously (see also figure 1). On the other hand, we can see that our method has no difficulty identifying object properties because it treats them as a set by leveraging slot attention and a matching procedure.

Table 2 shows the results for a particularly difficult training setting in which all of the objects are identical and have the same color, so the model cannot solely rely on color to separate objects, thus the matching has to be successful to enable disentanglement. It is needless to say that this scenario amounts to a non-injective $g$, and as can be seen, ResNet18 completely fails on any number of objects, whereas our method keeps achieving perfect disentanglement.

## 4. Discussion

In this work, we successfully extended recent techniques from causal representation learning Ahuja et al. (2022a,b) for fixed $d$—dimensional latent vectors with injective observation function $g(\cdot)$ to set-based latent vectors with non-injective $g(\cdot)$. To obtain such sets of representations, we adapted the slot attention architecture Locatello et al. (2020b) and showed perfect disentanglement results on a variety of synthetic scenarios. We believe this work takes a step towards closing the gap between theoretical results in causal identification and its real-world applications by considering the object-centricity of the natural scenes. That said, our modeling assumption of vector representations for object properties is restrictive, and we plan to relax this assumption in our future works. Moreover, our current framework also lacks any means to measure uncertainty about each property as we are not trying to model any distribution, whereas the distribution of the observations could well be described through a latent variable model based on object properties. Finally, our experiments should be extended to include 3D scenes, occlusions, different and more realistic backgrounds, and camera movements.

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

Extended Abstract Track

## Appendix A. Disentanglement Scores

Table 1: Disentanglement Scores for different methods when objects are not identical, i.e., have different colors.

| | LD | | | MCC | | |
|---|---|---|---|---|---|---|
| Number of objects | 2 | 3 | 4 | 2 | 3 | 4 |
| Ours | **0.96** | **0.90** | **0.87** | **0.98** | **0.94** | **0.93** |
| Slot Attention + RP | 0.14 | 0.07 | 0.05 | 0.16 | 0.15 | 0.08 |
| Slot Attention + PC | 0.16 | 0.18 | 0.10 | 0.18 | 0.30 | 0.18 |
| Slot Attention + LR | 0.44 | 0.24 | 0.17 | 0.47 | 0.31 | 0.24 |
| ConvNet (injective) | 0.99 | 0.97 | 0.98 | 0.91 | 0.95 | 0.97 |
| ConvNet (non-injective) | 0.25 | 0.12 | 0.07 | 0.39 | 0.27 | 0.21 |

Table 2: Disentanglement Scores for different methods when objects are identical, i.e., all have the same color.

| | LD | | | MCC | | |
|---|---|---|---|---|---|---|
| Number of objects | 2 | 3 | 4 | 2 | 3 | 4 |
| Ours | **0.99** | **0.94** | **0.92** | **0.99** | **0.97** | **0.96** |
| Slot Attention + RP | 0.19 | 0.07 | 0.06 | 0.28 | 0.15 | 0.12 |
| Slot Attention + PC | 0.40 | 0.20 | 0.13 | 0.36 | 0.25 | 0.17 |
| Slot Attention + LR | 0.43 | 0.25 | 0.16 | 0.50 | 0.36 | 0.23 |
| ConvNet (non-injective) | 0.31 | 0.15 | 0.07 | 0.44 | 0.32 | 0.24 |

## Appendix B. Experiments and Methods Details

We evaluate our method on a synthetic image dataset that allows us to carefully control various aspects of the environment, such as the number of objects, their sizes, shapes, colors, relative position, and dynamics. Such a controllable environment is an essential first step as it enables us to easily iterate and find the sources of non-identifiability for the proposed method. An example of our dataset is shown in figure 1. The general procedure in all experiments comprises a pair of images corresponding to consecutive steps $t, t + 1$ where there have been *some* changes, and the model could have *some* knowledge about these perturbations. As has been discussed so far, our goal is to identify the true latents at the object-level $z \in \mathbb{R}^p$ that give rise to model's observations $x$ by exploiting a weak supervision from sparse perturbations. These observations are produced via a data generation process (DGP) that consists of (1) a renderer or an observation function $\mathfrak{g}(\cdot)$ that translates object properties into a scene in the pixel space $x = \mathfrak{g}(z)$, and (2) a set of $k$ mechanisms $\mathcal{M} = \{m^1(\cdot), m^2(\cdot), ..., m^k(\cdot)\}$ that determine the transition dynamics in the

latent space $z$. Concretely, our synthetic DGP $\mathfrak{g}(\cdot)$ takes as input a set $\mathcal{Z}_t = \{\mathbf{z}_t^1, \mathbf{z}_t^2, \ldots, \mathbf{z}_t^n\}$ of $n$ object properties $\mathbf{z}_t^i$ where each object is specified as a $p-$dimensional vector ($p = 4$) of properties $\mathbf{z}_t^i = (x_i, y_i, c_i, s_i)_t$. $x, y, c, s$ denote the $x, y$ coordinates, color and shape, respectively. Based on object properties, they are each rendered and placed on a white background and then aggregated to produce $x_t$. To obtain the next observation $x_{t+1}$, we can select a subset of $m < n$ objects (denoted as the set $\mathcal{Z}' \subset \mathcal{Z}$) as targets that undergo perturbations determined by a subset of $m$ mechanisms $\mathcal{M}' \subset \mathcal{M}$ from the set of all possible mechanisms. The correspondence between the $m$ mechanisms and $m$ objects is determined by a random permutation $\pi_t^m$, i.e. $i = \pi_t^m[j]$ means that mechanism $i$ governs the transition dynamics of object $j$ if $\mathbf{z}_t^j \in \mathcal{Z}'$ (for objects that are not supposed to change from $t \to t+1$ a dummy mechanism with index $-1$ can be assumed which results in no change). A mechanism $m^i(\cdot) = (m_x^i(\cdot), m_y^i(\cdot), m_c^i(\cdot), m_s^i(\cdot))$ transforms the properties of the selected object $\mathbf{z}_t^j$ through the following relation:

$$\mathbf{z}_{t+1}^j = m^i(\mathbf{z}_t^j), \text{where} \quad x_{t+1}^j = m_x^i(x_t^j), \quad y_{t+1}^j = m_y^i(y_t^j), \tag{2}$$

$$c_{t+1}^j = m_c^i(c_t^j), \quad s_{t+1}^j = m_s^i(s_t^j) \tag{3}$$

This process is repeated until we obtain $\mathcal{Z}_{t+1}$, and through a similar rendering function we arrive at $x_{t+1}$. Note that the order of object representations in $\mathcal{Z}_t, \mathcal{Z}_{t+1}$ is the same and does not concern the model. The model receives $x_t, x_{t+1}$ along with some knowledge about the subset $\mathcal{M}'$ that caused the perturbations, and is tasked to jointly reconstruct the image at both $t, t+1$ as well as minimize an objective function in the latent space (see Equation 1). We show that this objective gives rise to disentangling the properties $x, y, c, s$ at the object-level. Note that the model is agnostic to the continuous or discrete nature of the true latents, and the objective regardless produces a disentangled representation. We can have a number of assumptions on the nature of the knowledge of perturbations that the model has access to.

## B.1. Matching

Perturbations (caused by the mechanisms) alter the properties of objects from $t \to t+1$ and the model has to figure out which mechanisms were applied to which balls to update its representations and minimize the latent loss (1). But recall that the model has no direct access to objects. It receives the observations at $t, t+1$ and encodes each of them to a set of slots $\mathcal{S}_t, \mathcal{S}_{t+1}$. These slots do not follow any fixed ordering, and moreover, there is no guarantee that each slot binds to exactly one unique object. Slots can also correspond to the background. Each perturbation $\delta_t^j$ changes the properties of some object $z_t^i$, so the model requires to find a pair of slots $(s_t^u, s_{t+1}^v)$ that are bound to object $z^i$ at $t$ and $t+1$, respectively. Once the model figures out such a matching, then the latent loss that results in disentanglement can be computed via the projections of these slots $\hat{z}$:

$$\mathcal{L}_z = \sum_{i=1}^m \|\hat{z}_t^i + \delta_t^{\pi_i^{\mathcal{M}'}[i]} - \hat{z}_{t+1}^i\|^2, \qquad \hat{z}_t^i = \hat{f}_z(s_t^u), \ \hat{z}_{t+1}^i = \hat{f}_z(s_t^v) \tag{4}$$

The problem of finding a correspondence between slot projections at $t, t+1$ and the perturbations is an instance of the 3-dimensional matching. We can use the following methods to solve this problem.

**Hungarian Matching**   If the changes from $t \to t+1$ are not dramatic, or the scene is not composed of exactly identical objects (same shape and color), then empirically we observe that more often than not, intializing the slots at $t, t+1$ results in sets of slots that preserve the ordering from $t \to t+1$. When this assumption is valid for many samples, our problem reduces to a bipartite matching of perturbations and slots for which the order does not change for two consecutive steps. This bipartite matching can be accomplished via the Hungarian algorithm Kuhn (1955). Concretely, the cost matrix required by the Hungarian algorithm $C_H$ is an $m \times |\mathcal{S}|$ matrix composed as the following:

$$C_H[i,j] = \|\hat{f}_z(s_t^j) + \delta_t^{\pi_t^{\mathcal{M}'}[i]} - \hat{f}_z(s_{t+1}^j)\|^2 \tag{5}$$

i.e., each row consists of the squared errors of applying one mechanism (corresponding to that row) to each slot (the columns). Hungarian algorithm finds an optimal assignment $\pi_t^*$ of slot projections and mechanisms such that $m^i(\cdot)$ is applied to $\text{MLP}(s_t^{\pi_t^*[i]})$ $\pi_t^*[i]$ for $i \in 1, \ldots, m$. Then the latent loss is computed as follows:

$$\mathcal{L}_z = \sum_{i=1}^m \|\hat{f}_z(s_t^{\pi_t^*[i]}) + \delta_t^i - \hat{f}_z(s_{t+1}^{\pi_t^*[i]})\|^2 \tag{6}$$

 Note that in equation B.1 the summation index runs over object indices, but in equation B.1 it runs over the mechanism indices (same range, but slightly different meaning.)

**Double Matching via Constrained Linear Programs (CLP)**   Although Hungarian matching can work in many situations, there exist some cases where the assumption of preserved order of slots from $t$ to $t+1$ does not hold anymore. For instance in situations where there exist a lot of symmetries (i.e., same color for all objects), then slots' binding to object will face a higher degree of randomness, and thus, using Hungarian matching would result in very noisy gradients that hinder convergence. Or when the perturbations are not very local, i.e. two relatively distant objects swap their positions from $t \to t+1$, then we can no longer assume that slots obtained at $t+1$ reflect the same binding to objects as slots that were obtained at $t$.

In such situations we resort to a more accurate matching scheme that significantly reduces the noise slot-object bindings and speeds up convergence drastically for these corner cases. This method deals with the more difficult problem of 3-dimensional matching, and uses slots at both $t, t+1$ to find the assignments, hence the name *double matching*. Recall that $\pi_t^s, \pi_t^{\mathcal{M}'}$ relate slots in $\mathcal{S}_t$ and mechanisms to the objects in $\mathcal{Z}_t$, respectively. Thus, the model at each step is required to jointly solve for these permutations at $t, t+1$ to minimize the following:

$$(\pi_t^s, \pi_{t+1}^s, \pi_t^{\mathcal{M}'})^* = \underset{\pi_t^s, \pi_{t+1}^s, \pi_t^{\mathcal{M}'}}{\arg\min} \sum_{i=1}^m \|\hat{f}_z(s_t^{\pi_t^s[i]}) + \delta_t^{\pi_t^{\mathcal{M}'}[i]} - \hat{f}_z(s_{t+1}^{\pi_{t+1}^s[i]})\|^2 \tag{7}$$

Notice that we are effectively finding the correspondence between perturbations $\mathcal{M}'$ and *pairs* of slots $(s_t^i, s_{t+1}^j)$ for $i, j \in [1 : |\mathcal{S}|]$, such that the pair of slots correspond to the same object as the one that is perturbed by the assigned mechanism. To find such assignments

we could construct an $m \times |S|^2$ cost matrix ($|S|^2$ denotes all the possible pairs of $(s_t^i, s_{t+1}^j)$) as follows:

$$C_{\text{CLP}}[i,j] = \|\hat{f}_z(s_t^{k_t(j)} + \delta_t^i - \hat{f}_z(s_{t+1}^{k_{t+1}(j)}))\|^2, \qquad i \in [1:m], j \in [1:|S|^2] \qquad (8)$$

$$k_t(j) = \lfloor j/|S| \rfloor \qquad (9)$$

$$k_{t+1}(j) = \text{mod}(j, |S|) \qquad (10)$$

However, the assignment cannot be recovered by Hungarian matching alone. The reason is that there are constraints that need to be satisfied for a matching in this scenario to be valid. Note that for each row $i$ (mechanism), the matched column index $j$ determines the pair of slots that correspond to the same object at $t, t+1$, which is perturbed by mechanism $i$. Such assignments have to satisfy the following constraints:

- Selected $j$s for all rows should be such that no slot at time $t$ is selected more than once as the first element of any slot pair, i.e. a slot cannot be the subject of two perturbations at any given $t$ because each object is affected by one and only one mechanism.

- Selected $j$s for all rows should be such that similarly, no slot at time $t+1$ is selected more than once as the second element of any slot pair, i.e. a slot cannot be the outcome of two perturbations at any given $t+1$ because again, each object is affected by one and only one mechanism.

Since any matching has to fulfill these constraints, it can no longer be treated as a simple bipartite matching solvable by the Hungarian algorithm. But we can still find an assignment as the solution of a constrained linear program (LP). We can define a binary $m \times |S|^2$ weight matrix $W$ such that when multiplied element-wise by $C_{\text{CLP}}$, masks all the entries that do not correspond to the matching by zero, and leaves the entries corresponding to the matching unchanged. Thus we can find the assignments simply by looking at the non-zero entries. So to summarize, the matching could be found by solving the following Constrained LP:

$$\text{minimize} \sum_{i=1}^{m} \sum_{j=1}^{|S|^2} W \odot C_{\text{CLP}} = \sum_{i=1}^{m} \sum_{j=1}^{|S|^2} C_{\text{masked}} \qquad (11)$$

$$\text{subject to} \qquad W[i,j] \in \{0,1\} \quad \forall i,j \qquad (12)$$

$$\sum_{i=1}^{m} \sum_{j \in \{k, k+|S|, k+2|S|, \dots\}} C_{\text{masked}}[i,j] = 1 \quad \forall k \in [1:|S|] \qquad (13)$$

$$\sum_{i=1}^{m} \sum_{j \in \{k|S|, k|S|+1, k|S|+2, \dots\}} C_{\text{masked}}[i,j] = 1 \quad \forall k \in [1:|S|] \qquad (14)$$

$$\sum_{j=1}^{|S|^2} C_{\text{masked}}[i,j] = 1 \quad \forall i \qquad (15)$$

Equations 13, 14 make sure the constraints mentioned above are satisfied, and the last constraint makes sure each mechanism is exactly assigned to one pair only. Although

solving this CLP provides an exact solution to our matching problem, we do not opt for a binarized $W$ as it will result in a mixed-integer CLP, which is NP-hard, and in practice would become intractable fairly quickly as the number of slots increases ($|S|^2$ dependence). Hence we will relax the constraint of equation 12 to $0 \leq W[i,j] \leq 1 \, \forall i,j$ to avoid any mixed-integer situation in the program. It is noteworthy to mention that although the relaxed CLP is significantly faster than the binarized version, it is still much slower than the Hungarian matching, despite our efforts to implement the constraints and the objective as parallel-friendly as possible. The bottleneck results from the constraints that we have introduced, but this is the price we need to pay to overcome those particularly hard cases with significant symmetries in the scene that otherwise could not be dealt with. As a matter of fact, even running Hungarian matching in those situations can still reasonably guide the latent representations toward disentanglement but it is for the evaluation of the predicted properties of *all* objects $\hat{z}$ against their true properties $z$ that we absolutely require double matching via CLPs.

**Disentanglement Metrics**    We need to compare $\hat{z}$, the projections of non-background slots, to the true latents $z$ of objects to measure the disentanglement of the properties in $\hat{z}$. We can evaluate identifiability of the learned representations either up to affine transformations or up to generalized permutations. These metrics are achieved by fitting a linear regression between $z, \hat{z}$ and reporting the coefficient of determination $R^2$, and using the mean correlation coefficient (MCC), respectively (Hyvarinen and Morioka, 2016, 2017).

