# OpenReview forum: "Object-centric causal representation learning"
_NeurIPS.cc/2022/Workshop/NeurReps — NeurReps 2022 Poster_

### Official Review · Reviewer_pAib · 2022-10-10
**Extension to nonlinear ICA method to deal with multiple possibly identical objects**

**Confidence:** 4
**Soundness:** 3
**Presentation:** 3
**Contribution:** 2
**Overall Rating:** 5

**Summary:**

The paper proposes a simple modification / combination of existing methods to deal with nonlinear ICA in settings were multiple objects may be confused.

**Questions:**

I wonder if a simple extension of the identifiably equivalence class to permutations (of subspaces) would suffice, theoretically, to account for the setting in this work?

Introduction (2nd paragraph): injectivity assumption (typo)

Data generating process: what are a sparse offsets? Are they one-hot, are they only sparse within each group of z?

Equation 1: are the deltas known? In that case, this isn‘t really blind sources separation (ICA/disentanglement) anymore, because of that supervision signal, right? How would these results / modelling approaches fare in settings were other forms of weak supervision (such as temporal statistics, https://arxiv.org/abs/2007.10930) are assumed?

**Limitations:**

Another interesting class of non-invertibility in image models is occlusions. I am wondering if any of the ideas from this work transfer to that?

There is much prior work on ICA when the mixing is non-invertible. Please do a bit of literature research and try to include some links to that literature. Much of recent ‚disentanglement‘ work ignores those theoretical links.

The problem is not that there are multiple objects, but that those multiple objects are, potentially, indistinguishable. If they were all, always, of different shapes or colours, then the fact alone that there are multiple objects would not lead to failures of injectivity. Please clarify.

**Recommended Decision:**

2: Borderline

**Relevance:**

2: Limited relevance

**Strengths And Weaknesses:**

One strength in the nonlinear ICA line of work is the theoretical results that stand in contrast to the trial-and-error approach ubiquitous elsewhere in machine learning. It would have been interesting, if the existing work went beyond combining existing models and had tried to see where existing identifiability proofs break without the invertibility assumption and which, potentially lighter, assumptions might replace it.

**Submission Track:**

Extended Abstract (4 Page)

---

### Official Review · Reviewer_8pdU · 2022-10-10
**Review of the paper "Object-centric causal representation learning".**

**Confidence:** 4
**Soundness:** 3
**Presentation:** 4
**Contribution:** 3
**Overall Rating:** 7

**Summary:**

In this work, the authors introduce a novel method to disentangle multiple objects in an image. The problem consists in correctly recognizing the generative factors of $k$ indistinguishable factors in a single image. This is the first work in which the injectivity assumption of the map from the generative factors to the input variables is relaxed.

As shown by the authors, current methods from disentanglement literature do not achieve substantial disentanglement when objects are indistinguishable, since they assume the injectivity of the map from factors to inputs. This becomes evident when an ordering of the objects cannot be established and the generative function is injective modulo a permutation of the $k$ sectors of the generative factors.

The authors adopt the slot-attention mechanism from Locatello et al. (2020b) and combine it with the Hungarian matching algorithm for sparse perturbations of the generative factors. In the case where such perturbations are not sparse, they propose a Constrained Linear program to match the representations of distinct objects.
For $k$ distinct objects, the proposed method achieves always good or comparable results with respect to other state-of-the-art methods and it outperforms the competitors in the case of indistinguishable objects, validating a successful disentanglement of the latent representations.

**Questions:**

From the text, it is not clear to me what are the methods compared in Tables 1 and 2. Can you explain to me what are Slot Attention + RP, Slot Attention + PC, and Slot Attention + LR? I think these acronyms should be introduced in the Appendixes.

Is there a specific reason to use LD and MCC in this context? Since the work relates to Causal Disentanglement, I would have chosen Interventional Robustness Score (IRS), in Suter et al. (2019).

What happens if objects overlap? Have you tested that regime?

**Limitations:**

The main limitation of this paper is the use of a toy dataset for the experimental investigation. I am looking forward to seeing how the method performs in other datasets. I already expressed my point of view on the chosen disentanglement metrics.

**Recommended Decision:**

3: Accept

**Relevance:**

3: Solid fit

**Strengths And Weaknesses:**

The paper addresses a difficult problem in causal representation learning, connected to object-centric learning. The presentation of the material is clear and the central points of the problem are well-explained in the text.  I found interesting their proposal to describe the objects as sets in the latent space, such that the vectorial representation should respect the order invariance.
The proposed method works effectively in this context and reveals a promising direction for object-centric learning. Overall, the quality of the paper is good. As a potential improvement, I suggest including other disentanglement metrics, such as DCI, MIG, or IRS (you can find a review of metrics here: https://arxiv.org/abs/2012.09276 ).


**Submission Track:**

Extended Abstract (4 Page)

---

### Official Review · Reviewer_S5Jj · 2022-10-15
**Interesting paper on representation learning with sets of objects, a bit dense in technical details**

**Confidence:** 2
**Soundness:** 3
**Presentation:** 2
**Contribution:** 3
**Overall Rating:** 7

**Summary:**

The paper describes an interesting approach for causal representation learning. It addresses the issue of injectivity when input scenes exhibit multiple objects. Then one would like that distinct parts of the latent space correspond to the distinct objects, however, this does not work when the latent is a vector space due to arbitrary permutations of the objects in the set (or vectorization of it). The paper solve this via an approach of slot-attention, which allows to sample an arbitrary random set of objects, which via an iterative scheme converge to the appropriate slots. A update matching scheme is proposed in order to map actions/perturbations to the corresponding objects. The paper presents a successful proof of concept of the results.

**Questions:**

Could you provide a high level description of the steps/general idea in causal representation learning through perturbations and matching.

**Limitations:**

The paper is clear in the limitations of the approach.

**Recommended Decision:**

3: Accept

**Relevance:**

3: Solid fit

**Strengths And Weaknesses:**

**Strenghts**

The paper addresses interesting (still open) challenges in causal representation learning. The authors provide a precise analysis of where current methods are limited when dealing with multiple objects and provides a sound solution to this problem.

**Weaknesses**

I found the paper very hard to read for someone with limit background in causal representation learning. The authors may assume to much prior knowledge of the reader.  That is, the whole idea of disentanglement learning via interventions or spare supervision is not well presented, but may be obvious for someone in the field. E.g., what are "sparse offsets", in what sense sparse because on page 3 the set \Delta seems to have an offset for all objects indexed with k? Also equation 1 could be further explained. I think the notion of "matching" is important, but hard to follow in the current paper. I assume this corresponds to matching permutations in the input and latent space, and training the encoder at the same time. On this topic, Hungarian algorithm is mentioned as well as "bipartite matching" for which it would be nice to add citations.

**Submission Track:**

Extended Abstract (4 Page)

---

### Decision · Program_Chairs · 2022-10-21

Accept (Poster)